



# Simulation of mixed-phase clouds with the ICON-LEM in the complex Arctic environment around Ny–Ålesund

Vera Schemann[1] and Kerstin Ebell[1]

[1]University of Cologne, Institute for Geophysics and Meteorology, Cologne, Germany

**Correspondence:** Vera Schemann (vera.schemann@uni-koeln.de)

**Abstract.**

Low-level mixed phase clouds have a substantial impact on the redistribution of radiative energy in the Arctic and are a potential driving factor for Arctic Amplification. To better understand the complex processes around mixed-phase clouds, a combination of long-term measurements and high-resolution modeling - which is able to resolve the relevant processes - is essential. In this study, we show the general feasibility of the new high-resolution model ICON-LEM to capture the general structure, type and timing of mixed-phase clouds at the Arctic site Ny-Ålesund and its potential and limitations for further detailed research. As a basic evaluation the model is confronted with data streams of single instruments including microwave radiometer and cloud radar, but also with value added products like the Cloudnet classification. The analysis is based on a 11-day long time period with selected periods being studied in more detail focusing on the representation of particular cloud processes, such as mixed-phase microphysics. In addition, targeted statistical evaluations against observational data sets are performed to assess i) how well the vertical structure of the clouds is represented and ii) how much information is added by higher resolutions. The results clearly demonstrate the advantage of high resolutions: in particular, with the highest model resolution of 75 m, the variability of liquid water path can be well captured. By comparing neighboring grid cells for different subdomains we also show the potential of the model to provide information on the representativity of single sites (as Ny-Ålesund) for a larger domain.

## 1 Introduction

The Arctic is warming at a higher rate than the global mean: the increase in the near-surface air temperature in the Arctic is more than twice as large as the observed increase in global mean temperature (Serreze and Barry, 2011; Wendisch et al., 2017). In order to better understand this phenomenon called Arctic amplification, many efforts are currently undertaken to pinpoint and quantify the related feedback mechanisms causing the enhanced climate change signal (e.g., Wendisch et al., 2017; Screen et al., 2018; Goosse et al., 2018). Low-level mixed-phase clouds are known to be one potential driver for Arctic Amplification and are very common in the Arctic (Shupe et al., 2008), but especially under Arctic conditions, many climate models struggle





to capture these clouds depending on their microphysics parameterization (Pithan et al., 2014) and to represent the boundary
layer structure due to low and strong inversions. To improve the relevant parameterizations in climate models, a better process
understanding and formulation is necessary and can be obtained by creating a synthesis of state-of-the-art observations and
high-resolution process modeling.

Concerning observations, enhanced measurements capabilities during specific campaigns can be of great value (e.g., Wendisch
et al., 2019; Tjernström et al., 2019; Shupe et al., 2006). In particular, the upcoming MOSAiC campaign (www.mosaic-
expedition.org) will provide for the first time continuous observations of the atmosphere, ice and ocean in the central Arctic
over a full year. While such campaigns provide a wealth of information from various instrumentation also for the inner Arctic,
they are always limited to a certain time period. However, in order to understand a changing climate, long-term measurements
are crucial. Such observations are available at the French-German Arctic research station AWIPEV at Ny-Ålesund, Svalbard
(78.925°N, 11.930°E). Ny-Ålesund is located at the south coast of the Kongsfjord and is surrounded by glaciers and moun-
tains which affect the local climate (Maturilli et al., 2013; Maturilli and Kayser, 2017). AWIPEV operates comprehensive and
state-of-the-art instruments in particular for thermodynamic, aerosol, trace gas and surface radiation observations where some
of the observations have been started more than 30 years ago enabling trend analyses (Maturilli et al., 2015; Maturilli and
Kayser, 2017). In 2016, a frequency-modulated continuous wave 94-GHz Doppler cloud radar of the University of Cologne
(Küchler et al., 2017) has been installed at AWIPEV providing highly temporally and vertically resolved cloud observations
and enabling now the analysis of microphyiscal processes of Arctic clouds in more detail at this site (Nomokonova et al., 2019).

The complex surrounding of Ny-Ålesund creates its own need for high-resolution simulations to be able to capture the
surface heterogeneities caused by the mixed surrounding of mountains, flat land, glacier and the fjord. Those conditions are
not feasible for the conventional idealized way to run Large-Eddy-Simulations with periodic boundary conditions and homo-
geneous surfaces. For this reason, we have applied the new Icosahedral Nonhydrostatic (ICON) Large-Eddy-Model (LEM)
for the first time in the Arctic. So far the model has been mainly applied over Germany (Heinze et al., 2017; Marke et al.,
2018) showing a reasonable representation of clouds and turbulence. Our main research question thus is, if the ICON-LEM
can reproduce the general structure of observed mixed-phase clouds identified by the observations at Ny-Ålesund by taking
into account the complex topography. Beyond the general classification, we also investigate how suitable the default micro-
physics and especially the cloud condensation nucleii (CCN) and the ice nucleii (IN) (Hande et al., 2016) are for the Arctic
regime. To investigate these questions, we picked a 10-day long time period, i.e. from June 14 to June 24, 2017, during the
ACLOUD/PASCAL campaigns (Wendisch et al., 2019), where in addition to the ground-based observations also aircraft-based
remote sensing and in-situ observations have been performed in the surroundings of Ny-Ålesund which will be used for further
analysis in future.

The advantage of a large-eddy-simulation is that we can simulate at temporal and spatial scales, which are comparable
to the observations. However, due to computational costs, we always have to find a balance between resolution and domain
size. A rather small and limited domain comes with the need of large-scale forcing to capture the general synoptic situation.
For this reason, forcing from numerical weather prediction models has to be applied to get information about the synoptic
situation and the large scales. As the large-scale models are struggling with the Arctic conditions and especially the inversion





strength (e.g. Pithan et al. (2014); Neggers et al.), forcing from two different weather prediction models (Integrated Forecasting

System (IFS) of the European Centre for Medium-range Weather forecasts (ECMWF) and the operational global ICON model

by the German Weather Service (DWD) (Zängl et al., 2015) have been used. The transition from large-scale forcing to the

high-resolution simulations is a well-known difficulty: an inconsistent forcing might introduce artifacts in the high-resolution

simulations or need very long spin-up times. This problem is mitigated by the new model suite ICON enabling a consistent

forcing throughout the whole model hierarchy. We can use the operational weather forecast simulations and force our large-

eddy-model with a consistent atmospheric state and a model setup, where only the respective parameterizations are switched

off or are replaced by a more suitable version (e.g. for turbulence). We will show how beneficial this feature of the ICON family

is and also look into the effect of increasing the resolution.

In this study, we demonstrate the general applicability of ICON-LEM in the Arctic, in particular in a place with a complex

topography, to evaluate and study clouds. After a description of the general model setup (sec. 2.1) and the observations used

(sec. 2.2), we will tackle our main research questions: Can the ICON-LEM capture the general structure of mixed-phase

clouds at Ny-Ålesund characterized by the Cloudnet classification (sec. 3)? Is the consistent forcing within one model family

beneficial (sec. 4.1)? Are the default microphysical parameterizations and a resolution of 75 m suitable for Arctic conditions

(sec. 5 and sec. 5.2)? And can we use high-resolution simulations to evaluate the representativity of point measurements at

complex locations (sec. 5.3)?

## 75  2  Setup

### 2.1  Model simulations

The Large-Eddy-Model of the ICON modeling system was developed during the HD(CP)$^2$ (High definition cloud and precip-

itation for advancing climate predictions) project and successfully tested and evaluated over Germany (Dipankar et al., 2015;

Heinze et al., 2017). In this study, we are showing the first application of this model in the Arctic facing the difficult terrain and

surface conditions around Ny-Ålesund. The setup consists of 4 different domains with one-way nesting. The largest domain

has a resolution of 600 m, a 3 s timestep, and a domain size of 110 km. The smallest domain has 75 m resolution on a 20 km

domain (Fig. 1), with a 3/8 s timestep. Due to the triangular grid, resolution in this context means edge length, which actually

gives a 2/3 higher resolution in using the traditional definition in which resolution is the root of the cell area. For the 11-day

period between 14 June and 24 June 2017, every day is simulated separately with a new initialization at 0 UTC. The simulations

are performed with the 2-moment microphysics scheme from Seifert and Beheng (2006) including 6 prognostic hydrometeors

(water vapor, cloud water, cloud ice, rain, snow, hail, graupel) and a Smagorinsky turbulence scheme (Dipankar et al., 2015).

The CCN and IN are described following a parameterization based on Hande et al. (2016). Due to the relatively small domain,

the large-scale forcing is very important to capture the general synoptic situation. The forcing is applied only at the boundaries

so that the flow can evolve and develop freely in the inner part of the domains. Nevertheless, the simulations depend on the

large-scale forcing and different forcing models can lead to different results. For this reason, we use two different models, i.e.

the IFS in a resolution of $0.1° \times 0.1°$ and the operational global ICON forecasts in the R3B7 (approx. 13 km) resolution, and





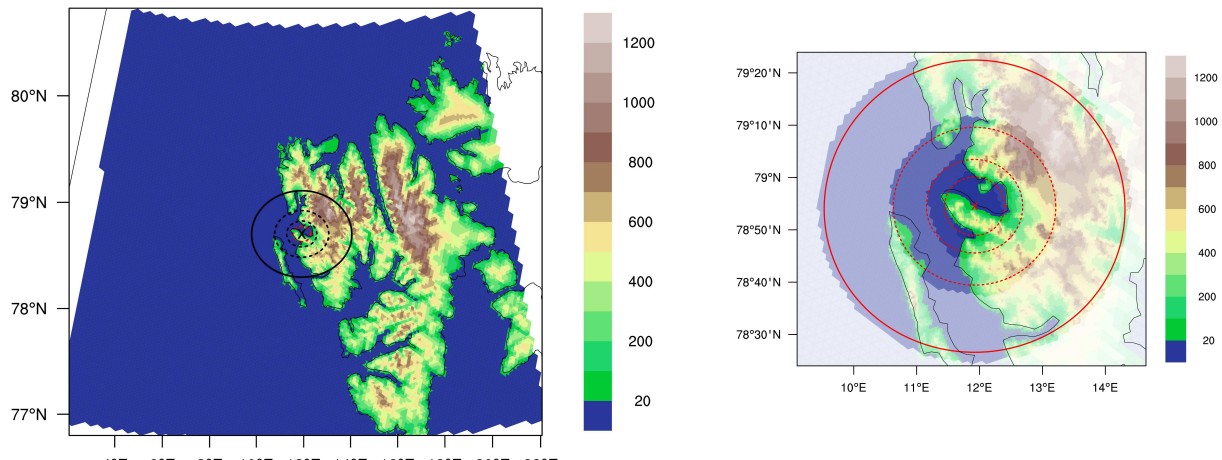

**Figure 1.** The topography, domain size and resolution around Ny-Ålesund for the 2 km ICON-NWP simulation (left) and the nested ICON-LEM simulations (right). The circles indicate the model domains for the 600 m, 300 m, 150 m, 75 m resolution model runs, respectively (from outer to inner circle) with corresponding domain sizes of approximately 110 km, 60 km, 35 km, and 25 km.

investigate the differences. A new forcing file is imported every hour for the IFS and every three hour for the ICON-global. While the IFS data profits from a rather high-resolution due to the fact that our study region is close to the pole, the ICON resolution stays due to the triangular grid structure at approx. 13 km, which is too coarse to force the ICON-LEM directly. For this

reason, we introduced an intermediate step with approximately 2 km resolution and adjusted parameterizations (ICON-NWP ) similar to the global simulations (see Fig. 1, left panel).

For the main part of the analysis, we use the so-called meteogram output, which is the column output at the grid-cell closest to the coordinates of the Ny-Ålesund measurements. The output is written every 9 seconds, which brings it close to the temporal resolution of the observational data sets. Due to the included topography and open boundaries, we expect the column

to be representative for the conditions in Ny-Ålesund and by this provide a better estimate than traditional quantities like the domain mean or variance. Additionally, the weather is often large-scale driven and further affected by local topography and surface conditions which are accounted for in ICON-LEM. Nevertheless, those point-to-point comparisons can cause further uncertainties for model-observation comparisons, e.g. by missing clouds or certain structures which might be represented in neighbouring cells. For this reason, we also included the two-dimensional output of liquid water path (LWP) in our analysis,

which has been written on 10-minute time intervals.

## 2.2   Observational data set

The model simulations are compared to observations performed at the atmospheric observatory of the Arctic French-German research station AWIPEV at Ny-Ålesund. In this study, we use information from microwave radiometer and cloud radar observations and from a synergistic classification product.





The 94-GHz Doppler cloud radar of the University of Cologne provides vertical profiles of cloud radar reflectivity factor $Z$, Doppler velocity and spectral width up to a height of about 12 km. In this study, we make use of the cloud radar reflectivity profiles which have been brought to a common 30-s and 20-m temporal and height grid, respectively. In addition to the active component, the cloud radar also has a passive channel at 89 GHz. The measured brightness temperatures at 89 GHz were used to retrieve LWP as described in the next paragraph.

Information on integrated water vapor (IWV) and liquid water path (LWP) was taken from the 14-channel microwave radiometer (MWR) HATPRO at AWIPEV. Details on the HATPRO retrievals can be found in Nomokonova et al. (2019). The 1-s MWR measurements were averaged to a common 9-s temporal grid similar to the ICON-LEM model output. Since the HATPRO was not measuring between June 21 and June 24 and had also a few data gaps on other days, we took additional LWP information from a statistical retrieval based on the additional passive 89 GHz channel of the cloud radar. For this LWP 120 retrieval, we combined 89 GHz brightness temperature measurements with IWV information from GPS. In the case that the HATPRO LWP is not available but the LWP from the 89 GHz retrieval is, the latter one is used instead, resulting in a combined, best-estimate data set for LWP. For the analysis of the power spectrum, continuous data are crucial. We thus divided the time series into 6-h intervals and excluded those intervals from the analysis which still suffered from data gaps.

While the cloud radar reflectivity profiles provide information on the vertical occurrence of hydrometeors, more detailed in125 formation on hydrometeor type is provided by the Cloudnet target classification product (Illingworth et al., 2007; Nomokonova et al., 2019). For this classification, each radar height bin is classified with respect to the occurrence of cloud liquid droplets, ice, melting ice and drizzle/rain and finally, the profiles of cloud radar reflectivity, Doppler velocity, and of ceilometer attenuated backscatter are combined with numerical weather prediction data. The resulting classification profiles have the same temporal (30 s) and vertical (20 m) resolution as the cloud radar measurements.

**3 Basic evaluation of clouds**

We use the Cloudnet target classification as a first and easy assessment of the general representation of structure, type and timing of the modeled versus the observed clouds. The Cloudnet classification also provides an impression of the changing meteorological conditions e.g. frontal passages or occurrence of low-level mixed-phase clouds. The classification of the model output is based on a threshold of $10^{-8}$ kg/kg for the hydrometeors and shows a reasonable agreement (Fig. 2) to the Cloudnet 135 classification described in the previous section. The general situation is mostly captured by the models and also type, structure and timing are represented well. Nevertheless, some differences especially in the duration of mixed-phase clouds can be spotted immediately. With regards to resolution, the 2 km resolution already shows a reasonable agreement, even though it has a tendency to generate more precipitation. However, it shows that the large-eddy-simulation benefits from a good representation by the numerical weather prediction forcing data.

The variability of the atmospheric conditions during the 11 days has already been indicated by the Cloudnet classification but can also seen in the timeseries of IWV and LWP (Fig. 3). For the IWV (top), both resolutions follow nicely the observed general trend. As the IWV is mainly dominated by larger scales, hardly any difference can be seen between the lowest (600 m)

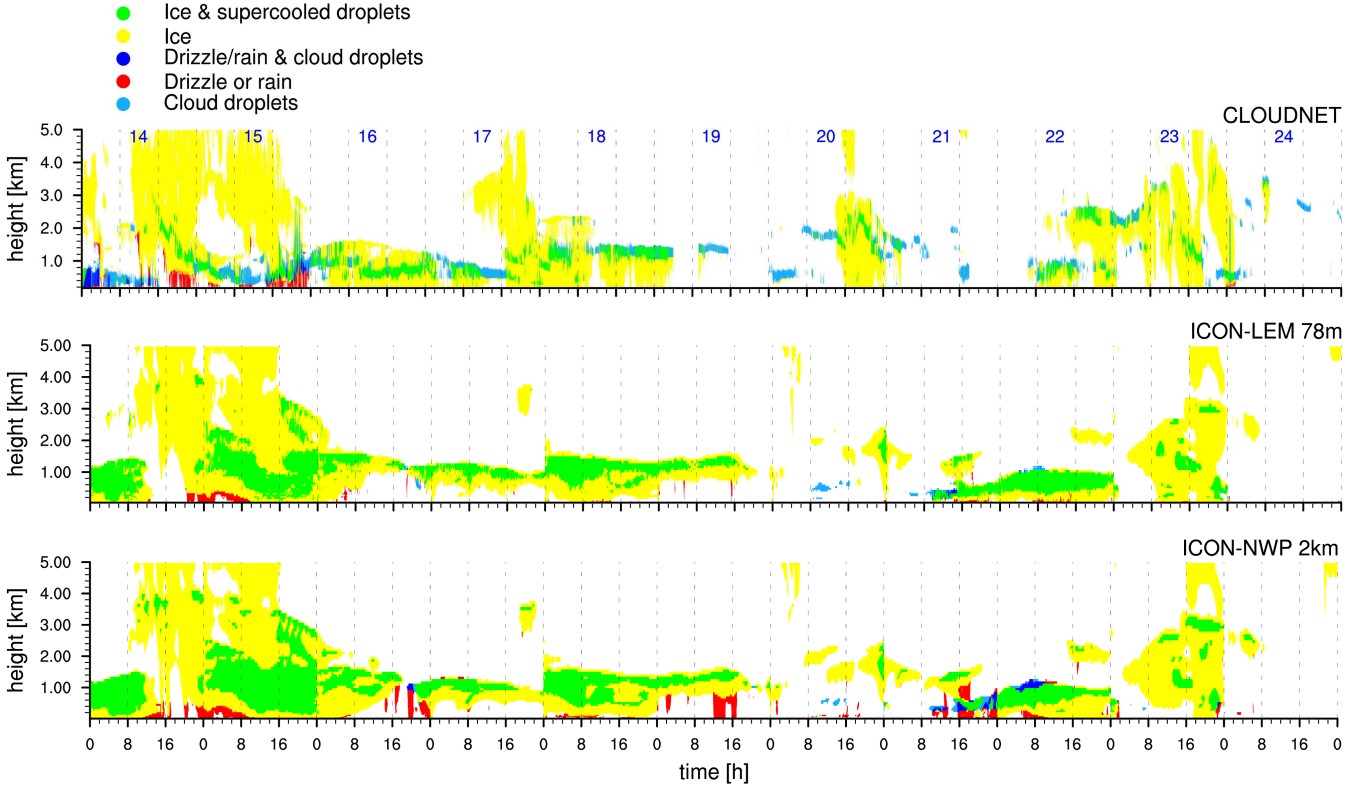

**Figure 2.** Hydrometeor classification for the whole time series for the observations (Cloudnet classification; top), the ICON-LEM 75 m (middle), and the ICON-NWP 2 km forcing data.

and highest (75 m) resolution. Strong gradients often occur at 0 UTC, which are due to the new initialization at 0 UTC. The model output between 0 and 6 UTC should be treated with caution as it includes the model spinup but is shown here for

completeness. For the LWP (bottom), the models also capture most of the clouds and variability, even though some clouds are missing, which could already be seen in the classification (Fig. 2). For the LWP, the two different ICON-LEM resolutions deviate from each other which is analyzed and shown in more detail in section 5.2.

## 4   Zooming in: Case study 23 June 2017

The previous section showed that the 2 km forcing data already capture the general structure, type and timing of the clouds

during the analyzed time period. The very similar classification time series of ICON-LEM and ICON-NWP indicate that the representation in the ICON-LEM is strongly influenced by the forcing data. We will thus investigate the forcing dependency in more detail by focusing on 23 June 2017 which reveals a complicated cloud structure with a very thin mixed-phase cloud and several liquid layers.

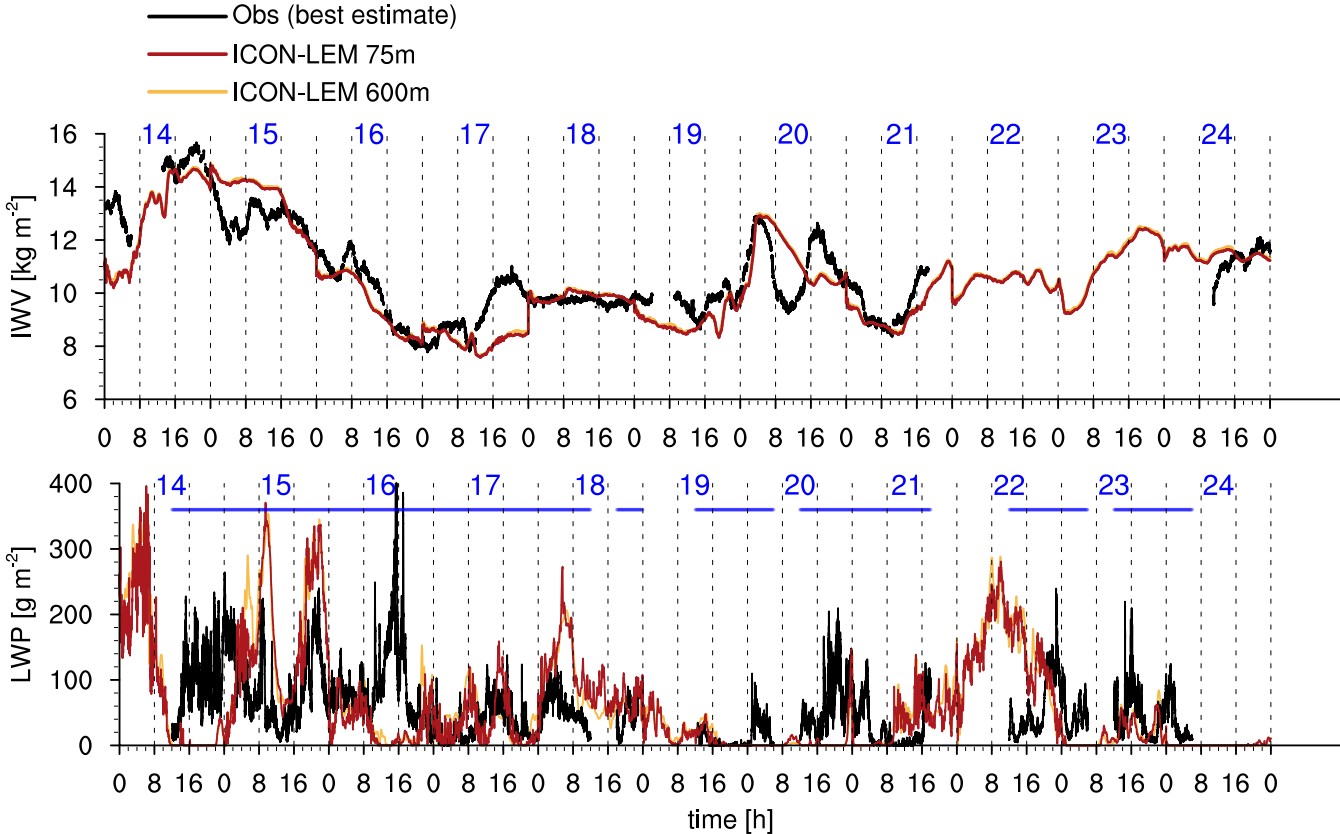

**Figure 3.** Time series of observed (black) and of ICON-LEM simulated (600 m: yellow, 75 m: red) IWV (top) and LWP (bottom). Blue lines show data availability of the observations.

### 4.1 Forcing dependency

The 23 June 2017 case is a very strong example of the impact of different forcing models on the representation of mixed-phase clouds in the ICON-LEM. Figure 4 shows the hydrometeor classification of this day – again for the observations and the ICON-LEM but also for the 75 m output of the ICON-LEM forced by IFS data. The bottom panels in Fig. 4 show the forcing data itself. While the ice cloud is represented in all forcing data, it is evaporated immediately and not fully recovered in the ICON-LEM simulation forced with the IFS data. The reason for the sudden evaporation is probably a different set of
parameterizations and the relation between subgrid-scale (ice) clouds and the mean state in the models. The transition to a different state representation in the ICON model leads to a mismatch. The ICON-LEM with a 75 m resolution and forced by the ICON model chain captures the cloud situation with low- and mid-level mixed-phase clouds and higher ice cloud in the afternoon much better than the one forced with the IFS. The mixed-phase clouds around 2 km height in the beginning of the day are not captured by ICON-LEM which might still be connected to the spin-up time of the model. This example shows the





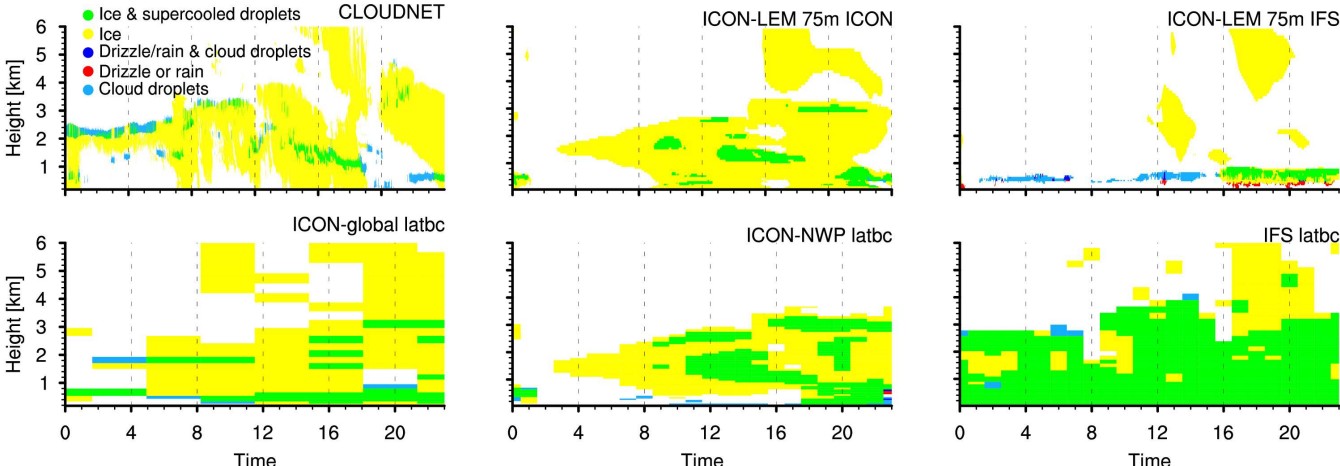

**Figure 4.** Classification for the case study of 23 June 2017 showing the observations (top, left) the ICON-LEM results at 75 m resolution with forcing from the ICON family (top middle) and forcing from the IFS (top right). The respective forcing data from the operational global ICON (ICON-global; bottom left), the ICON at 2 km resolution (ICON-NWP; bottom middle) and from the IFS (bottom right) are also shown.

importance of applying consistent forcing data, which is possible with the new ICON suite that can simulate at scales ranging from climate to large-eddy-resolving.

## 4.2   Vertical structure of the clouds

While the general structure is captured in the ICON-LEM simulation with the ICON forcing, we are also interested in the composition of the clouds and the dominating microphysical properties and processes. One the one hand, cloud properties
which have been retrieved from observations could be directly compared to the model results. However, retrieval algorithms applied to measurements may induce large uncertainties in this comparison. On the other hand, the modeled mixed-phase cloud properties can be evaluated by comparing observed cloud radar reflectivities with forward simulated reflectivities based on the ICON-LEM model output. Figure 5 shows the observed reflectivities as well as the ones from the ICON-LEM which were forward simulated with the Passive and Active Microwave TRAnsfer model (PAMTRA; (Maahn et al., 2015)). Radar
reflectivity depend on both, hydrometeor concentration and size. We consistently find, that the simulated reflectivities are lower than the observed ones. This underestimation might indicate that the ICON-LEM clouds consist of too small particles. One possible explanation could be the limitation of the applied CCN and IN parameterization. Another issue could be in the description of growing processes for the ice clouds.



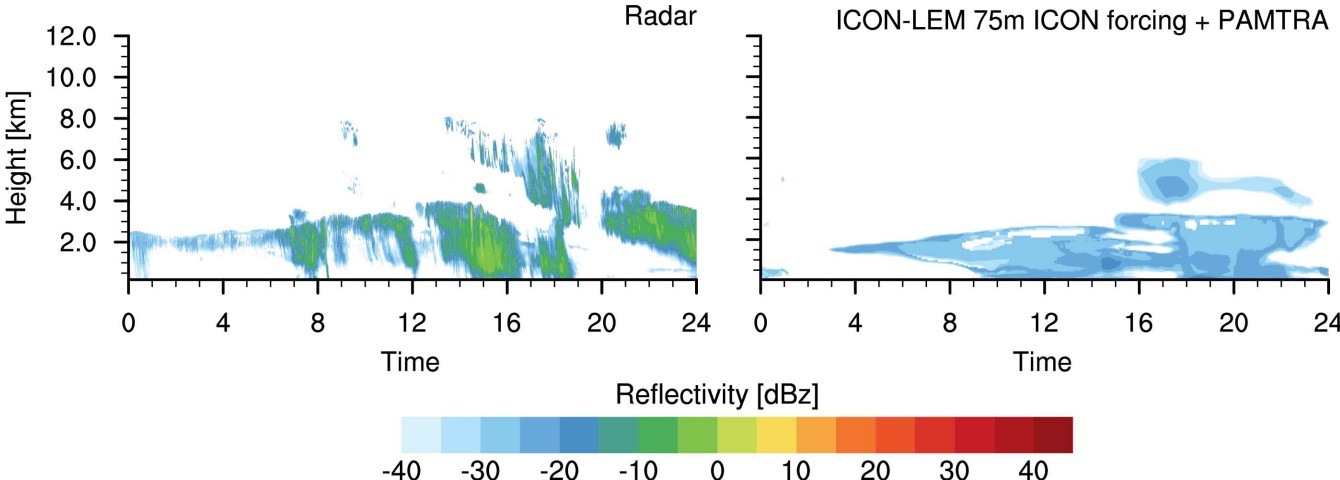

**Figure 5.** Time series of observed (left) and simulated cloud radar reflectivity based on the ICON-LEM with 75 m resolution (right) for the case study of 23 June 2017.

## 5 Statistical evaluation

While case studies allow us to investigate certain situations in detail, they might not be representative for the general behaviour. In this section, we use all 11 days to tackle the question how well the microphysical composition of the clouds is represented in the simulations and how much information can be added by higher resolutions.

### 5.1 Reflectivity distribution

As the ice cloud at the 23 June 2017 is very thin and challenging for the model (as seen in the previous section), the underesti-
mation of the radar reflectivity might not be representative for the general model behaviour. We thus compare the observed and simulated reflectivities for all 11 days in a 2D histogram (Fig. 6). The frequencies are based on the total number of possible data points (e.g. 9600 for the model output). With this approach, the distributions provide information about the total frequency, not only within one height. Interestingly, in the model, radar reflectivities are basically confined to values between -32 and -20 dBZ with two distinct peaks around -29 dBZ and -23 dBz. Up to 1.5 km height, observed radar reflectivities range between -36 to
-24 dBZ. Such a higher occurrence of radar reflectivities can also be seen in the model. The histogram confirms the results of the case study that the simulated reflectivities tend to be too low compared to the observed ones. This becomes even more clear for the clouds around 3 km height, where the observed frequencies shift towards higher values, while the simulated ones stay close at -30 dBZ. However, the observed and the simulated reflectivities cover in principal the same range indicating the potential to reach a better representation by refined microphysical parameterizations. The too high occurrence of simulated low
radar reflectivities and the two distinct peaks go along with very small and persistent ice water contents (not shown) and might be due to overestimated ice nucleation particles (IN) and cloud condensation particles (CCN). In order to better explain the differences between simulated and observed radar reflectivities more detailed sensitivity studies are needed disentangling the





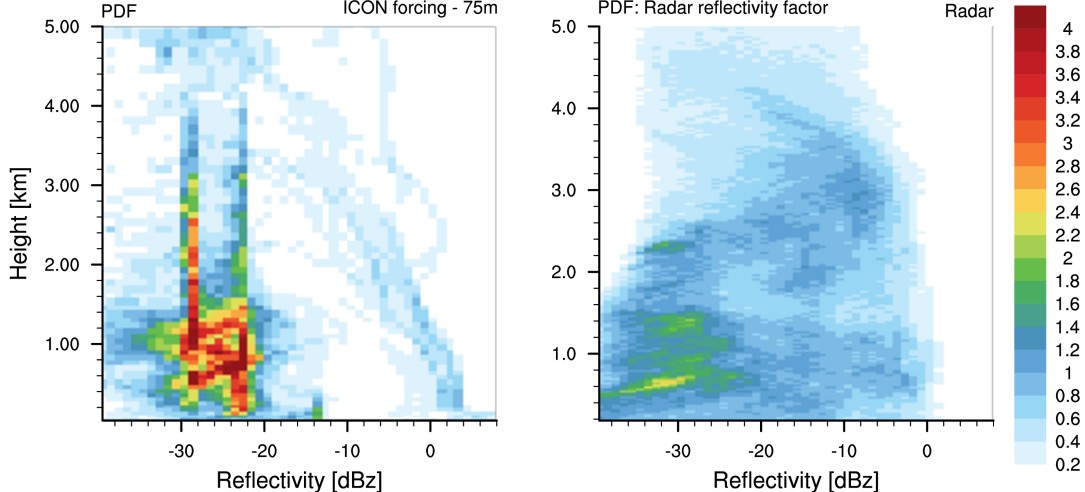

**Figure 6.** 2D histogram for simulated (left) and observed (right) radar reflectivities for all 11 days (14-24 June 2017). The PDF (%) of each height is based on the maximal possible number of datapoints (normalized).

effects of CCN, IN, and microphysical processes. This will be part of future research with upcoming long-term measurements of IN and CCN.

## 5.2 Resolution dependency

When looking at the general representation of the cloud structure as given by the hydrometeor classification (Fig. 2), the difference between different resolutions of ICON was rather small. In this section, a more detailed analysis of the impact of the different resolution will be presented. The analysis is performed on the meteogram output, which has an output frequency of 9 s and approaches the temporal scale of the observations. Being able to compare observations and model output at similar scales, is one of the key argument for high-resolution simulations. Still, the questions remain on which scales variability is resolved by the model and which resolution is needed to resolve the main part of the observed variability. Figure 7 qualitatively shows how the structure of resolved features gets finer and more detailed with increased resolution, i.e. 600 m, 300 m, 150 m, and 75 m, from the outer to the inner circles. To quantify this first impression, we calculated the power spectrum of the integrated quantities (LWP, IWV and IWP). Due to data availability, we divided the days into four parts each 6 hour long and picked the sequences with full data availability (see section 2.2 and Fig. 3 for details). For each 6 hour time series, we calculated the power density spectrum $P(f_i)$ for the frequencies $F = [\Delta f, \ldots, \frac{1}{2\Delta}]$, where $\Delta$ is the smallest time interval of the dataset. As the forcing data is produced at a different time frequency, we divided each power density spectrum by the bin width $\Delta f$ to be independent of the bin width. As the total amount of variance differs between the simulations and the observations, and we are interested in the scaling behavior, the power density spectrum has been normalized by the total variance

$$p(f_i) = \frac{P(f_i)}{\Delta f} \frac{1}{\sum_{f_j \in F} P(f_j)} \tag{1}$$

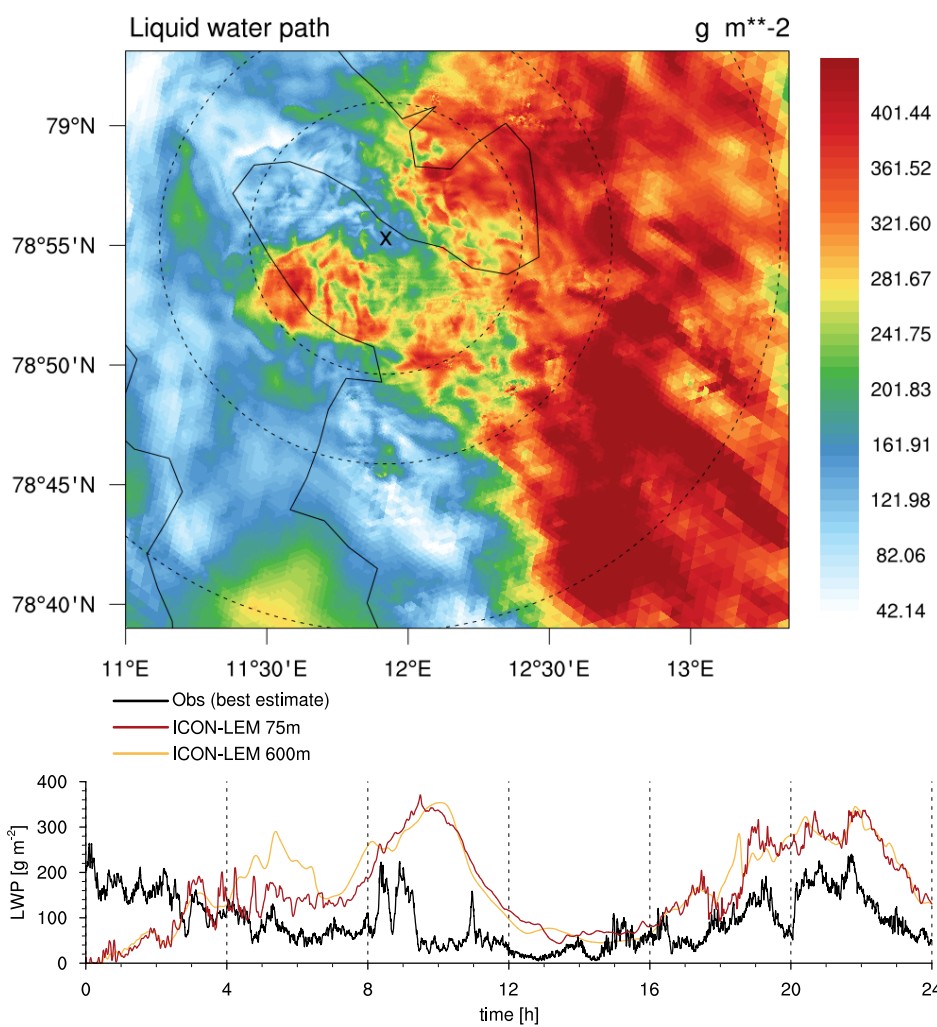

**Figure 7.** Snapshot of the LWP at 15 June 2017 6:30 pm showing an overlay of all four different resolutions (right) and LWP time series for the the same day showing the best estimate of the observations and the coarsest (600 m) as well as finest (75 m) ICON-LEM resolution.





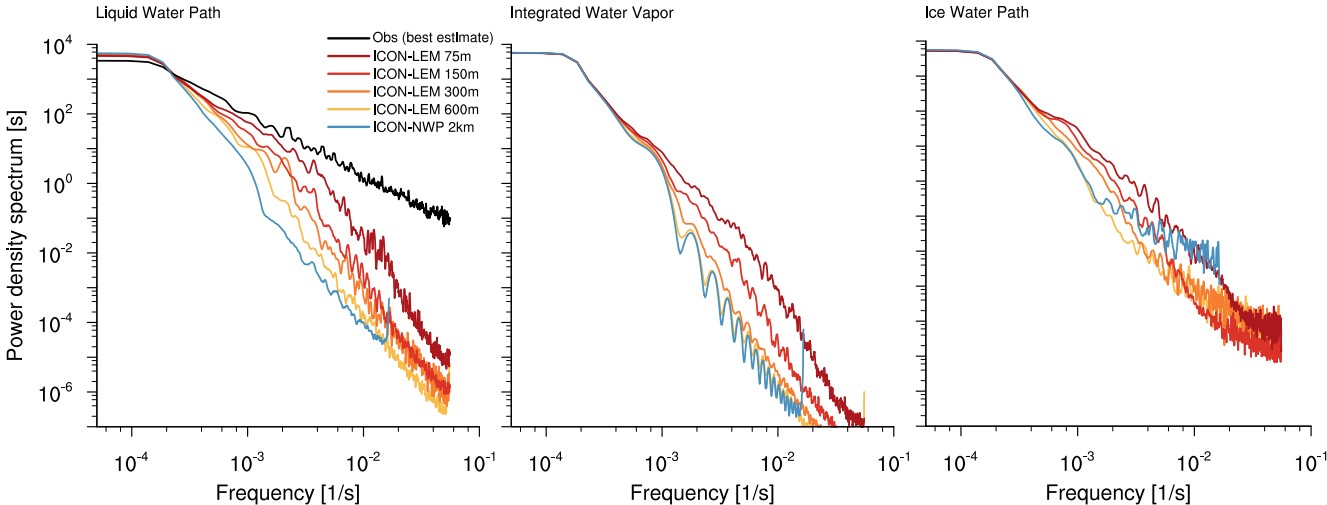

**Figure 8.** Power density spectrum for LWP (left), IWV (middle) and IWP (right) for the forcing ICON-NWP (blue) and the ICON-LEM for the four different resolutions ranging from 600 m (yellow) to 75 m (dark red). For LWP also the observations are shown (black). The spectrum is averaged over several 6 hour time slots excluding 0-6 hour every morning due to spin up and time slots without available observations (see Fig. 3).

Figure 8 shows the normalized power density spectrum for the observations, the four different resolutions of the ICON-LEM and the 2 km ICON-NWP. Since continuous highly resolved (i.e. here 9 s) IWV and IWP observations are not available, only the retrieved LWP is shown in the analysis. While especially for the forcing data the resolved variability is dominated by the large scales, we see an increase of variability at smaller scales with the four ICON-LEM resolutions. Especially for the 75 m

resolution, the model approaches the variability of the observations. While, of course, the observations also contain a certain amount of noise – which might dominate the small scale variability – it could be an interesting experiment in the future to test even higher model resolutions. For now we see a clear improvement in the representation of the small scale variability by increasing the resolution from 600 m to 75 m.

For IWV and IWP (see Fig. 8) the behavior is very similar. The highest resolution resolves more energy at smaller scales

than the coarser resolutions. While the IWV spectrum decays very quickly with smaller scales, the IWP spectrum decays later, more similar to the LWP. This shows that the IWV is mainly large-scale driven while for IWP and LWP smaller scales and fluctuations play a more important role, even though they are still partly unresolved in the model as the comparison with the observations for the LWP clearly shows.

## 5.3 Testing the representativity

The complex terrain around Ny-Ålesund is also a further challenge for point-to-point comparisons between the model and the observations. A slight mismatch in the location of the compared model column might already lead to a completely different environment compared to the observational site. For this reason, we are interested in the question of how strongly neighbouring





columns vary, and if a point-to-point comparison is meaningful at all. This question is a subaspect of the longstanding question of how representative measurements at a supersite like Ny-Ålesund are for Arctic regions in general. The model allows us to

slowly approach these questions. To show the potential, we take the coarse 2 km ICON simulations and select four different small sub-regions with different environments (Fig. 9, left): the coastal site Ny-Ålesund, sea-ice, open water and mixed conditions (where both, sea ice and open water occur). For each of these places, we pick 10 neighbouring cells and compare the PDF of LWP of the 9 outer ones with the original cell in the center to estimate the local variability. The PDFs are compared based on their mean value and the Hellinger distance, which can be used to compare also discrete distributions:

$$H(C,N) = \frac{1}{\sqrt{2}} || \sqrt{C} - \sqrt{N} ||_2 = \frac{1}{\sqrt{2}} \sqrt{\sum_{i=1}^{k} (\sqrt{c_i} - \sqrt{n_i})^2},$$

with $0 \leq H(C,N) \leq 1$, and $C$ being the PDF of the original cell and $N$ of a single neighbouring cell ($N \in \{N_1, \ldots, N_9\}$). For identical distributions, $H$ equals zero. The maximum distance $H = 1$ is achieved for completely disjunct PDFs, which do not overlap.

We see, that - as could be expected - the differences of the PDFs are smallest for open water and sea ice conditions. Under

mixed conditions close to the fjord, we already see larger differences among the 9 grid points as indicated by a slightly larger Hellinger distance and larger variations in the mean value of LWP. Clearly, we find the largest Hellinger distance for Ny-Ålesund as well as the largest differences in the LWP mean values. For a resolution of approximately 2 km this is certainly expected as the neighbouring grid cells might be characterized by very different surfaces - e.g. coast, mountains, fjord, and glaciers. The larger Hellinger distance and the increased variability of the mean LWP around Ny-Ålesund compared to the other

points shows that the local features are captured by the model. Certainly the point-to-point comparison has to be done carefully for Ny-Ålesund, but the fact that the local features can be seen by the model is promising for a reasonable representation of the column output. In future, measurements from a scanning MWR can be used to also describe the spatial variability of LWP from the observations and set this one then directly into context to the model LWP fields. At the moment we can only evaluate the model output at Ny-Ålesund and the small differences over water or sea ice might also be due to a too simplistic representation

within the model. Upcoming campaigns (e.g. MOSAiC) will enable us to also evaluate the model under varying conditions, as over sea ice. Then the model can be evaluated and tested at observational anchor sites but will provide information on larger domains – e.g. covering the whole Arctic – enabling the comparison of Ny-Ålesund to a larger region.

## 6  Conclusions

Low-level mixed-phase clouds, which are very common in the Arctic (Shupe et al., 2008), have a substantial impact on the

redistribution of radiative energy in the Arctic and are driven by very complex processes (Morrison et al., 2012). Thus, their representation is a challenge for today's climate models (Pithan et al., 2014). To improve the respective parameterizations and our process-level understanding, we need tools like a combination of detailed as well as long-term observations and high-resolution models. In this study, we presented the first simulations with the new ICON-LEM under Arctic conditions and the complex terrain of Ny-Ålesund. We demonstrated its capability to capture the general structure, type and timing of the observed





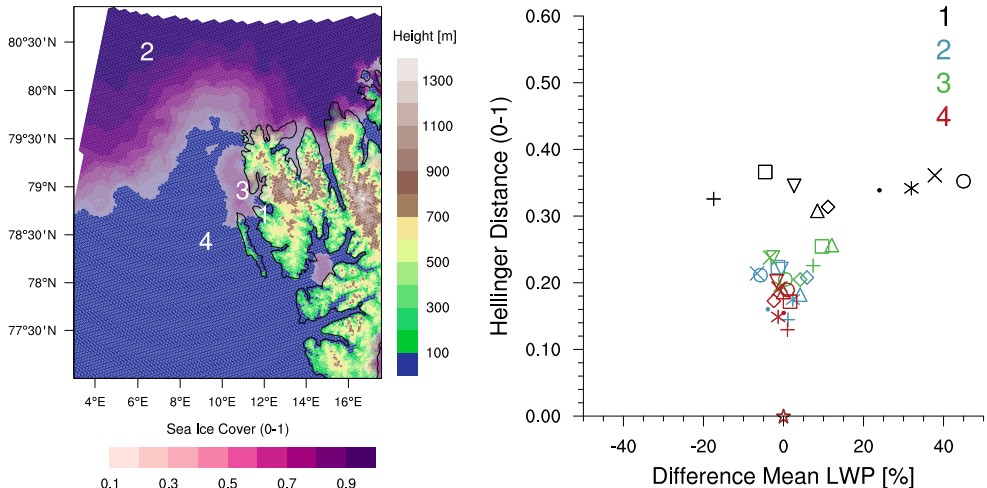

**Figure 9.** Left: Location of the four 10-grid point subregions used for the representativity analysis: land (1), sea ice (2), mixed conditions (sea ice and open water) (3), open water (4). Right: Hellinger distance and differences in mean LWP of the LWP PDFs of the surrounding grid points compared to the central point for the four subregions. Values have been calculates for each day and then averaged over all 11 days.

mixed-phase clouds. By analyzing 11 days during the ACLOUD campaign, we showed the potential for more detailed studies focusing on the composition of mixed-phase clouds and the dominating microphysical processes. To understand microphyiscal processes, it is important to bring the simulations and the observations on a similar scale, which can only be done with high resolutions. While the overall structure of the observed clouds is already captured in general by the large-scale forcing, we showed large differences, especially for the variability of the LWP, between the different resolutions. While certainly also a

75 m resolution is still too coarse for convergence and higher resolutions might still add more information, we could already see a distinct information gain from the lower (2 km, 600 m, 300 m, 150 m) to the highest ICON resolution (75 m).

    For the high-resolution model, we chose a rather small domain, which increases the dependency on the large-scale forcing and by that the induced uncertainty. Based on a case study, we demonstrated the effect of using two different forcing data sets as well as the benefit of having a consistent forcing with a similar model. This consistent forcing can be achieved with the new

ICON model suite, which allows simulations on scales reaching from the global and climate scales to Large-Eddy-resolving scales. Even though the parameterizations still have to be exchanged or switched off, the definition of the atmospheric state and the dynamics stay the same, which is a great advantage and reduces the uncertainty in the high-resolution simulations as well as the necessary spin-up time.

    We found a persistent underestimation of radar reflectivity in the simulations, which hints at too small particles and could

be due to the applied CCN and IN background forcing. Especially in the clean Arctic regime, the sensitivity on CCN and IN might be higher than in the mid-latitudes and should be investigated in more detail also in the ICON-LEM setup.



One long-standing question is the representativity of point measurements in general and especially at complex locations like Ny-Ålesund. We presented a first attempt to show the potential of high-resolution modeling to tackle these questions by offering a four-dimensional context to the point measurements. For a more detail study, it is necessary to evaluate the model

also under different conditions like sea-ice in the central Arctic to be able to conclude on the performance of the model under different conditions and draw conclusions for the representativity. Upcoming campaigns like MOSAiC will offer the necessary observational data sets and open up new possibilities for the synthesis of high-resolution modeling and observations in the Arctic.

The presented combination of long-term as well as state-of-the-art observations and the novel ICON model suite - including

the ICON-LEM - can lead to an improved process understanding and therefore to a better representation also in the large-scale models, which will allow us to investigate feedback and climate mechanisms related to Arctic Amplification in more detail.

*Data availability.*   ICON-LEM model data are available at the tape archive of the German Climate Computing Center (DKRZ; https://www.dkrz.de/up/systems/hpss/hpss); one needs to register at DKRZ to get a user account. We will also make the data available via Swift (https://www.dkrz.de/up/systems/swift) on request. The Cloudnet data are available at the Cloudnet website (http://devcloudnet.fmi.fi/,

Cloudnet, 2018). The IWV data of the MWR are available at PANGAEA (https://doi.pangaea.de/10.1594/PANGAEA.902142).

*Author contributions.*   VS performed the model simulations and the respective analysis of the model output. KE processed the observational data and supported the analysis. Both prepared the manuscript.

*Competing interests.*   The authors declare that they have no conflict of interest.

*Acknowledgements.*   We gratefully acknowledge the funding by the Deutsche Forschungsgemeinschaft (DFG, German Research Foundation)

– project number 268020496 – TRR 172, within the Transregional Collaborative Research Center "ArctiC Amplification: Climate Relevant Atmospheric and SurfaCe Processes, and Feedback Mechanisms (AC)[3], and the computing time granted on the supercomputer MISTRAL at Deutsches Klimarechenzentrum GmbH (DKRZ) through it's Scientific Steering Committee (WLA). We also thank Mario Mech for performing the forward simulations with PAMTRA for the model output and Christoph Ritter and Marion Maturilli of the Alfred Wegener Institute, Helmholtz Centre for Polar and Marine Research, Potsdam, for providing MWR and ceilometer data used in the Cloudnet algorithms, respec-

tively. We also thank AWIPEV for hosting the cloud radar of University of Cologne and the AWIPEV team for helping us in the instrument operation.



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
