# Peer review of "Simulation of mixed-phase clouds with the ICON-LEM in the complex Arctic environment around Ny-Ålesund"

_Atmospheric Chemistry and Physics, 2019_

## Referee Comment (RC1) · Anonymous Referee #1 · 2 Aug 2019

General comments

This manuscript uses observations from a high-latitude site in the Arctic to determine how well high-resolution models can represent clouds in this region, especially mixed-phase clouds. The major objective is to assess the improvement that higher-resolution modelling provides for capturing mixed-phase clouds, and also investigate the spatial representativity of vertical profile measurements from a supersite, particularly in these regions with complex orography and heterogeneous surfaces. The methodology is relevant with the results of clear importance to the community, and the comparison between different forcing datasets is also of interest, although this aspect is not ex-

plored deeply in this study. This manuscript is almost ready for publication, with a few technical aspects to correct and a few minor points to consider

Minor comments

It is clear that this a preliminary study highlighting some important features rather than an exhaustive study. There is an attempt to provide quantitative rather than merely qualitative measures for assessing the improvement that higher resolution provides, however some more detail could be presented. For example, Fig. 6 shows the output of PAMTRA compared to observations and there seems to be some issues with the microphysical parametrizations used. How about checking how well bulk quantities such as cloud fraction or IWC compare? These should not be so dependent on the CCN/IN parametrizations used in PAMTRA and would show whether the model at least has the bulk properties correct.

Figure 8 shows that all model resolutions show a similar gradient in the LWP power density spectrum for high frequencies, which is presumably due to the numerical dissipation (with the gradient depending on the scheme). Can the effective grid-scale resolution be determined from these plots? I.e. what approximate multiple of the grid resolution are scales resolved? This is also useful to know when discussing representativity of individual columns.

Technical comments

Title and elsewhere: Suggest replacing 'the ICON-LEM' with 'ICON-LEM' or 'the ICON-LEM model'.

Line 2: Add hyphen to 'mixed phase'.

Line 18: Replace 'currently' with 'currently being'.

Line 28: Replace 'measurements' with 'measurement'.

Line 39: Replace 'has been' with 'was'.

[Figure]

Line 40: Remove 'now'.

Line 42, 102: Replace 'those' with 'these'.

Lines 46-48: Suggest rephrasing this sentence.

Line 49: The 'parametrizations of CCN and IN'.

Line 59: Reference missing year.

Lines 67, 72, 81, Fig. 1 and elsewhere: To be clear, state 'horizontal resolution', especially in the setup section, and also specify the typical vertical resolution of the model at the altitudes the mixed-phase clouds are present.

Line 79: Replace 'we are showing' with 'we show'.

Line 89: Replace 'inner part of the domains' with 'inner domain'.

Line 94: Replace 'stays due' with 'stays constant due'.

Lines 110-114: Please include a reference to this instrument and the settings employed (e.g Nomokonova et al., 2019).

Line 131: Replace 'as a first and easy' with 'for the initial'.

Line 135: Suggest rephrasing slightly; e.g. 'benefits from a a good representation of the large-scale atmospheric forcing in the NWP data'.

Line 143: Replace 'initialization' with 'model initialization'.

Line 175: Remove both commas on this line.

Line 180: Insert 'model' behaviour.

Line 181: Insert 'of' how well.

Line 184: Replace 'at the' with 'on'.

Line 187: Replace ', not only within one height' with 'at all heights simultaneously'

[Figure]

Line 196: Insert 'number concentrations', i.e. 'ice nucleation particle (IN) and cloud condensation particle (CCN) number concentrations'.

Line 207: Replace 'increased' with 'increasing'.

Line 237: Remove comma (both sea ice and ..).

Lines 284-286: This sentence can be removed.

Acknowledgements: Include reference to ACTRIS for providing the Cloudnet output.

Line 302: Replace 'it's' with 'its'.

Figure 2: Centre panel title states 78 m whereas caption states 75 m.

Figure 4: I assume that 'latbc' in the panel titles refers to 'lateral boundary conditions'. This could be added in the figure caption.

Figure 7: For clarity, please explain in the caption that the concentric dashed circles represent the outler limits of the various domains (presumably), the coastline is indicated by the solid black line Ma, and also state that X marks the location of Ny Ålesund. The first panel is (upper) not (right), at least in this preprint.

Figure 9: Make it clear in the caption that the different symbols in the right panel refer to the surrounding grid points in each sub region. It's also challenging to identify subregion 1 in the left panel.

---

## Referee Comment (RC2) · Anonymous Referee #2 · 19 Sep 2019

A feasibility study is performed to see whether the ICON- large eddy model (LEM) realistically simulates a mixed phase cloud event at Ny-Alesund during June 2017. Spatial complexity provides a stiff modeling challenge. This is the first time the model has been used in the Arctic, having primarily been applied in Germany. High resolution in space and time are advocated as being needed for simulations of cloud liquid water, as demonstrated by Figure 8. By contrast, Figure 7 did not prove compelling.

Most issues are left for the future, such as the modeled low radar reflectivity being caused by shortcomings in the microphysics parameterization. Because the symbols were not identified in the right side plot, I didn't get much out of the representativity

analysis in Figure 9.

Overall, the paper reads well and should be published after modest changes. Can the authors provide other examples of LEMs being used to simulate Arctic mixed phase clouds so the relative skill of this model can be evaluated?

---

## Author Comment (AC1) · 1 Nov 2019

**Answer to anonymous reviewer 1**

**General comments**

**This manuscript uses observations from a high-latitude site in the Arctic to determine how well high-resolution models can represent clouds in this region, especially mixed-phase clouds. The major objective is to assess the improvement that higher-resolution modelling provides for capturing mixed-phase clouds, and also investigate the spatial representativity of vertical profile**

[Figure]

**measurements from a supersite, particularly in these regions with complex orography and heterogeneous surfaces. The methodology is relevant with the results of clear importance to the community, and the comparison between different forcing datasets is also of interest, although this aspect is not explored deeply in this study. This manuscript is almost ready for publication, with a few technical aspects to correct and a few minor points to consider.**

We thank the reviewer for his or her time and the careful reading and comments. We hope, that we could answer all comments and questions in a satisfying way.

**Minor comments**

**It is clear that this a preliminary study highlighting some important features rather than an exhaustive study. There is an attempt to provide quantitative rather than merely qualitative measures for assessing the improvement that higher resolution provides, however some more detail could be presented. For example, Fig. 6 shows the output of PAMTRA compared to observations and there seems to be some issues with the microphysical parametrizations used. How about checking how well bulk quantities such as cloud fraction or IWC compare? These should not be so dependent on the CCN/IN parametrizations used in PAMTRA and would show whether the model at least has the bulk properties correct.**

Cloud fraction/occurrence is basically shown in Fig. 2. Comparing bulk properties like IWC is more difficult since retrieval algorithms have to be applied to the measurements before. These retrieval algorithms often go along with large uncertainties hampering a firm assessment of the model results. A standard IWC product is based on the method by Hogan et al. (2006). They found that uncertainties of the IWC retrieval

differ for different temperature ranges and are estimated to be from -50% to +100% for temperatures below -40°C and ranging from -33% to 50% for temperatures above -20°C. We thus decided to restrict this first assessment to the observational space. In future, a more detailed evaluation of the modeled ice/snow characteristics will be performed by exploiting also the information from polarimetric and dual-frequency cloud radar measurements.

Hogan, R., M. Mittermaier, and A. Illingworth, 2006: The retrieval of ice water content from radar reflectivity factor and temperature and its use in evaluating a mesoscale model. J. Geophys. Res., 45, 301–317.

**Figure 8 shows that all model resolutions show a similar gradient in the LWP powerdensity spectrum for high frequencies, which is presumably due to the numerical dissipation (with the gradient depending on the scheme). Can the effective grid-scale resolution be determined from these plots? I.e. what approximate multiple of the grid resolution are scales resolved? This is also useful to know when discussing representativity of individual columns.**

For the estimation of the effective grid resolution, a spectrum of the turbulent kinetic energy or vertical velocity is way more recommended, than any humidity spectrum, which is also influenced by other processes. We didn't investigate the effective resolution or the energy spectrum, but this has been done for the model before in Heinze et al. 2017, where they say, that the estimated effective resolution is approximately 8 times the horizontal resolution.

Heinze, R., et al.,: Large-eddy simulations over Germany using ICON: a comprehensive evaluation, Quarterly Journal of the Royal Meteorological Society, 143, 69–100, https://doi.org/10.1002/qj.2947, https://rmets.onlinelibrary.wiley.com/doi/abs/10.1002/qj.2947, 2017

**Technical comments**

**Title and elsewhere: Suggest replacing 'the ICON-LEM' with 'ICON-LEM' or 'the ICON-LEM model'.**
As the ICON-LEM is not a name but an acronym, which already contains the "model", we decided to stick with "the ICON-LEM".

**Line 2: Add hyphen to 'mixed phase'.**
Done.

**Line 18: Replace 'currently' with 'currently being'.**
Done.

**Line 28: Replace 'measurements' with 'measurement'.**
Done.

**Line 39: Replace 'has been' with 'was'.**
Done

**Line 40: Remove 'now'.**
Done

**Line 42, 102: Replace 'those' with 'these'.**
Done

**Lines 46-48: Suggest rephrasing this sentence.** We rephrased the sentence to: "Our main research question thus is, if the ICON-LEM can reproduce the general structure of the observed mixed-phase clouds at Ny-Ålesund by taking into account the complex topography."

**Line 49: The 'parametrizations of CCN and IN'.**
Done

**Line 59: Reference missing year.**
Updated

**Lines 67, 72, 81, Fig. 1 and elsewhere: To be clear, state 'horizontal resolution',especially in the setup section, and also specify the typical vertical resolution of the model at the altitudes the mixed-phase clouds are present.**
We added the term "horizontal" several times in the manuscript and also included some information on the vertical resolution. We still left the pure 'resolution' at some parts, as of course with the horizontal resolution also the temporal resolution is changed (as mentioned in the setup) which also influences especially the small scale variability e.g. for LWP.

**Line 79: Replace 'we are showing' with 'we show'.**
Done

**Line 89: Replace 'inner part of the domains' with 'inner domain'.**
Not only the inner domain is meant, but the inner part of each domain, as each domain is forced at the boundaries (even though this effect is of course strongest at the outermost domain).

**Line 94: Replace 'stays due' with 'stays constant due'.**
Done

**Lines 110-114: Please include a reference to this instrument and the settings employed(e.g Nomokonova et al., 2019).**
We added the references Küchler et al. (2017) and Nomokonova et al. (2019).

**Line 131: Replace 'as a first and easy' with 'for the initial'.**
Done

**Line 135: Suggest rephrasing slightly; e.g. 'benefits from a good representation of the large-scale atmospheric forcing in the NWP data'.**
We rephrased the sentence accordingly.

**Line 143: Replace 'initialization' with 'model initialization'.**
Done

**Line 175: Remove both commas on this line.**
Done

**Line 180: Insert 'model' behaviour.**
Done

**Line 181: Insert 'of' how well.**
Done

**Line 184: Replace 'at the' with 'on'.**
Done

**Line 187: Replace ', not only within one height' with 'at all heights simultaneously'**
Done

**Line 196: Insert 'number concentrations', i.e. 'ice nucleation particle (IN) and cloud condensation particle (CCN) number concentrations'.**
Done

**Line 207: Replace 'increased' with 'increasing'.**
Done

**Line 237: Remove comma (both sea ice and ..).**
Done

**Lines 284-286: This sentence can be removed.**
We rephrased the sentence instead: "In future, it is necessary to evaluate the model also under different conditions like sea-ice in the central Arctic."

**Acknowledgements: Include reference to ACTRIS for providing the Cloudnet output.**

Done

**Line 302: Replace 'it's' with 'its'.**
Done

**Figure 2: Centre panel title states 78 m whereas caption states 75 m.**
Thank you very much for this hint! We corrected the panel title.

**Figure 4: I assume that 'latbc' in the panel titles refers to 'lateral boundary conditions'. This could be added in the figure caption.**
Thank you, we added the explanation to the caption.

**Figure 7: For clarity, please explain in the caption that the concentric dashed circles represent the outler limits of the various domains (presumably), the coastline is indicated by the solid black line Ma, and also state that X marks the location of Ny Ålesund. The first panel is (upper) not (right), at least in this preprint.**
Thank you very much! We adjusted the caption.

**Figure 9: Make it clear in the caption that the different symbols in the right panel refer to the surrounding grid points in each sub region. It's also challenging to identify subregion 1 in the left panel.**
Thank you very much, we adjusted the caption.

**Answer to anonymous reviewer 2**

**A feasibility study is performed to see whether the ICON- large eddy model (LEM) realistically simulates a mixed phase cloud event at Ny-Alesund during June 2017. Spatial complexity provides a stiff modeling challenge. This is the first time the model has been used in the Arctic, having primarily been applied in Germany. High resolution in space and time are advocated as being needed for simulations of cloud liquid water, as demonstrated by Figure 8. By contrast, Figure 7 did not prove compelling. Most issues are left for the future, such as the modeled low radar reflectivity being caused by shortcomings in the microphysics parameterization. Because the symbols were not identified in the right side plot, I didn't get much out of the representativity analysis in Figure 9. Overall, the paper reads well and should be published after modest changes. Can the authors provide other examples of LEMs being used to simulate Arctic mixed phase clouds so the relative skill of this model can be evaluated**

We thank the reviewer for the time and the positive feedback. We would like to clarify some points. We agree with the reviewer, that Figure 8 is way more compelling to show the resolution dependency. Nevertheless, the power spectrum is a rather complex and combined product and we decided to include the Figure 7 to have an easier example of the resolution dependency. We also extended the caption of Figure 9 and hope, that it is now easier to understand.

Even though there are several LES simulations on mixed-phase clouds, it is difficult to use these to evaluate the relativ skill of the ICON-LEM. Previous studies are based on very idealized cases and setups, while our approach needed the new model developments which allow to include topography and lateral boundary conditions in order to simulate real cases and day-to-day variability of mixed-phase clouds in a complex environment like Ny-Ålesund. Nevertheless, we included three more references to idealized LES studies to provide more context to the reader.

**Additional changes**

Caption of Figure 1: "(m)" added

Line 50:
"10-day" replaced by "11-day"

Line 92:
"every three hour" replaced by "every three hours"

Line 169:
"One" replaced by "On"

We added further references in the text:
Line 40:
Gierens et al. (2019)

Gierens, R., Kneifel, S., Shupe, M. D., Ebell, K., Maturilli, M., and Löhnert, U.: Low-level mixed-phase clouds in a complex Arctic environment, Atmospheric Chemistry and Physics Discussions, 2019, 1–37, https://doi.org/10.5194/acp-2019-610, https://www.atmos-chem-phys-discuss.net/acp-2019-610/, 2019

Line 260:
Miller et al. (2015), Ebell et al. (2019)

Ebell, K., Nomokonova, T., Maturilli, M., and Ritter, C.: Radiative effect of clouds at Ny-Ålesund, Svalbard, as inferred from ground-based remote sensing observations, Journal of Applied Meteorology and Climatology, https://doi.org/10.1175/JAMC-D-19-0080.1, 2019

Miller, N. B., Shupe, M. D., Cox, C. J., Walden, V. P., Turner, D. D., and Steffen, K.: Cloud Radiative Forcing at Summit, Greenland, Journal of Climate, 28, 6267–6280, https://doi.org/10.1175/JCLI-D-15-0076.1, 2015